# Associations of modern initial antiretroviral therapy regimens with all-cause mortality in people living with HIV in resource-limited settings: a retrospective multicenter cohort study in China

Xinsheng Wu [1,2,19], Guohui Wu [3,19], Ping Ma [4,5,19], Rugang Wang[6,19], Linghua Li[7,19], Yuanyi Chen[1,2], Junjie Xu[8], Yuwei Li [1,2], Quanmin Li[7], Yuecheng Yang[9], Lijing Wang[10], Xiaoli Xin[11], Ying Qiao[12], Gengfeng Fu [13] ✉, Xiaojie Huang[14] ✉, Bin Su [14] ✉, Tong Zhang[14] ✉, Hui Wang[15] ✉ & Huachun Zou[16,17,18] ✉

Despite the proven virological advantages, there remains some controversy regarding whether first-line integrase strand transfer inhibitors (INSTIs)-based antiretroviral therapy (ART) contributes to reducing mortality of people living with HIV (PLHIV) in clinical practice. Here we report a retrospective study comparing all-cause mortality among PLHIV in China who were on different initial ART regimens (nevirapine, efavirenz, dolutegravir, lopinavir, and others [including darunavir, raltegravie, elvitegravir and rilpivirine]) between 2017 and 2019. A total of 41,018 individuals were included across China, representing 21.3% of newly reported HIV/AIDS cases collectively in the country during this period. Only the differences in all-cause mortality of PLHIV between the efavirenz group and the nevirapine group, the dolutegravir group and the nevirapine group, and the lopinavir group and the nevirapine group, were observed in China. After stratifying the cause of mortality, we found that the differences in mortality between initial ART regimens were mainly observed in AIDS-related mortality.

Integrase strand transfer inhibitor (INSTI) has been introduced and widely recommended as the third drug in modern first-line treatment for people living with HIV (PLHIV) since its first approved in 2007[1,2]. Previous studies have demonstrated its exceptional efficacy and safety[3–5]. However, not all countries are capable of timely promoting the use of INSTIs due to varying levels of development, economic status, and healthcare.

In China, the National Free Antiretroviral Treatment Program (NFATP) was launched in 2002 and has resulted in a significantly reduction in mortality rates for PLHIV[6–10]. In June 2016, China implemented a treat-all policy, which grants access for all PLHIV to antiretroviral therapy (ART) regardless of CD4+ T-cell count[11]. Before 2009, the ART regimens recommended were zidovudine (AZT) or stavudine + lamivudine (3TC) + nevirapine (NVP) or efavirenz (EFV)[12].

A full list of affiliations appears at the end of the paper. ✉e-mail: fugf@jscdc.cn; huangxiaojie78@ccmu.edu.cn; binsu@ccmu.edu.cn; zt_doc@ccmu.edu.cn; huiwang98@szsy.sustech.edu.cn; hzou@kirby.unsw.edu.au

The Chinese Guidelines for Diagnosis and Treatment of HIV/AIDS 2011 incorporated the tenofovir-based ART regimens recommended by the WHO for resource-limited settings, consisting of a combination of tenofovir disproxil fumarate (TDF) + 3TC + EFV, which was also made available through the NFATP[12,13]. Over the past years, the NFATP-sponsored ART regimens have undergone continuous evaluation and improvement[14]. INSTIs, new generations of protease inhibitors (PIs), among others were introduced and promoted nationally, which substantially expanding access to ART initiation and switching for PLHIV in China[14]. Since 2015, raltegravir (RAL) and dolutegravir (DTG) have been used in PLHIV in China[15–18]. In 2018, single-regimen pills containing emtricitabine (FTC)/tenofovir alafenamide fumarate (TAF) and INSTI gained widespread useage[15–18].

However, a recent real-world study from high-income countries found little evidence of differences in mortality rates between most modern first-line regimens[19]. In routine clinical care, decisions could be influenced by more factors than in a randomized trial, including side effects, adherence, and regimen tolerability, among others. For example, several studies have reported side effects of INSTI, including weight gain[20,21], obesity[22,23] and cardiovascular disease[24] of PLHIV.

Evidence from low- and middle-income countries (LMICs) in routine clinical care would provide a clearer picture, as regimens commonly used in high-income countries are difficult to access or afford in resource-limited settings. Nonnucleoside reverse-transcriptase (NNRTI)- and PI-based ART regimens were used the most commonly in LMICs[25,26]. It has been reported in some LMICs that the majority of PLHIV on ART receive an NNRTI-based ART regimen, while those who meet a failure with NNRTI-based regimens are mostly switched to a PI-based regimen[25,26]. Moreover, long-term INSTI-based prescriptions are expensive in China. In 2015, two INSTIs received regulatory approval and began to be used in PLHIV in China[15–18]. However, the landscape of fully reimbursed ART in China before 2021 included two NNRTIs (EFV and NVP), and a single PI (lopinavir [LPV]); along with several nucleotide reverse transcriptase inhibitors (NRTIs). All INSTIs and most PIs were not available free of charge before 2021[14]. Elucidating the effects on mortality of different ART regimens in practice is therefore crucial to the global ambition of achieving the UNAIDS 95-95-95 targets by 2030. In this study, we conducted a retrospective multicentre cohort study to provide evidence for decision-making in routine clinical care in resource-limited settings.

## Results

A total of 41,018 individuals who started ART between January 2017 and September 2019 were included in the analyses. Overall, males accounted for 84.6% of the total sample, and the median age was 37 years (interquartile range [IQR] 28–52). 36,838 (89.8%) patients initiated NNRTI-based regimen, 1574 (3.8%) initiated INSTI-based regimen, and 2606 (6.4%) initiated PI-based regimen. 2777 (6.8%) switched regimen, 1132 (2.8%) loss to follow-up (LTFU), 14,366 (39.5%) achieved a CD4 > 500 from the 36,396 patients with initial CD4 < 500, and 656 (1.6%; 393 AIDS-related death and 263 non-AIDS-related death) died during 65,578 person-years of follow-up (median 1.6 years [IQR 1.0–2.2]). In the analysis of cumulative incidence for CD4 improvement, only 36,396 individuals with initial CD4 < 500 were included. Detailed characteristics of individuals included were presented in Table 1.

The proportion of PLHIV who started ART with EFV-based ART regimen decreased from 2017 to 2019, whereas the proportion with DTG increased in the same period (Fig. 1A). Specifically, 15,196 (87.4%) patients initiated EFV and 212 (1.2%) initial DTG as the third drugs in their regimen in 2017, but this changed to 6211 (81.2%) and 498 (6.5%) in 2019. In addition, the proportion of PLHIV who initiated ART with others uncommon drugs (including darunavir [DRV], RAL, elvitegravir [EVG] and rilpivirine [RPV]) as the third drugs increased from 132 (0.8%) in 2017 to 300 (3.9%) in 2019. The proportion of PLHIV deaths by initial ART regimen from 2017 to 2019 were shown in Fig. 1B.

There were different risks of incidence for ART switch, LTFU, CD4 improvement, and death (Fig. 2). In the univariate Poisson regression model, compared to patients who initiated NVP based ART regimen, patients who initiated EFV (incidence rate ratio [IRR] 0.45, 0.32–0.62; Table S1) and LPV (0.57, 0.41-0.79) based ART regimens had significantly lower rates of incidence of ART switch, while patients who initiated DTG (2.07, 1.56–2.75) and others (2.62, 1.91–3.59) based ART regimens had significantly higher rates of incidence of ART switch. There was no significant difference in the incidence rate of LTFU between initial ART regimens. Compared to patients who initiated NVP based ART regimen, patients who initiated EFV (1.46, 1.21–1.76), DTG (1.96, 1.58–2.42), LPV (1.72, 1.42–2.08) and others (1.77, 1.39–2.26) based ART regimens had significantly higher rates of incidence of CD4 improvement; patients who initiated EFV (0.44, 0.22–0.85), DTG (0.23, 0.07–0.74), and LPV (0.23, 0.09–0.55) based ART regimens had significantly lower rates of incidence of all-cause mortality. Results for different causes of death were shown in Table S1.

In the multivariate Poisson regression model, age at 35–49 years (adjusted incidence rate ratio [aIRR] 3.04, 2.37–3.90; Reference: 18–24 years; Table 2 and Fig. 2), 50+ years (7.37, 5.77–9.42), location in the Southwestern (2.86, 2.08–3.93; Reference: Northern China) and Eastern China (2.34, 1.68–3.26), and >30 days to ART initiation (2.00, 1.50–2.67; Reference: same-day initiation) were risk factors for increased incidence of all-cause mortality; whereas female (0.54, 0.49–0.60; Reference: male), homosexual transmission (0.50, 0.44–0.58; Reference: heterosexual), location in the Southern China (0.22, 0.11–0.45; Reference: Northern China), and initial ART in 2018 (0.86, 0.80–0.93; Reference: 2017) were protective factors. Compared to patients who initiated NVP based ART regimen, patients who initiated EFV (0.62, 0.51–0.76), DTG (0.43, 0.18–0.98), and LPV (0.54, 0.36–0.80) based ART regimens had significantly lower rates of incidence of all-cause mortality. After stratified by cause of death, compared to patients who initiated NVP based ART regimen, patients who initiated EFV (0.48, 0.39–0.60), DTG (0.28, 0.11–0.72), and LPV (0.22, 0.13–0.37) based ART regimens had significantly lower rates of incidence of AIDS-related mortality; patients who initiated other uncommon drugs (6.53, 2.05–20.80) based ART regimens had significantly higher rates of incidence of non-AIDS-related mortality. Results for ART switch, LTFU, and CD4 improvement were shown in Table S2.

For each third drug comparison, differences of all-cause mortality between the EFV group and the NVP group (aIRR 0.64, 0.53–0.79; Table 3), the DTG group and the NVP group (0.27, 0.11–0.67), and the LPV group and the NVP group (0.49, 0.33–0.75), were significant. After stratified by cause of death, differences of AIDS-related mortality between the EFV group and the NVP group (aIRR 0.49, 0.40–0.62), the DTG group and the NVP group (0.17, 0.08–0.40), the LPV group and the NVP group (0.20, 0.12–0.33), and the LPV group and the EFV group (0.46, 0.29–0.74), were significant; differences of non-AIDS-related mortality between the others group and the NVP group (aIRR 7.74, 1.03–58.17), and the others group and the EFV group (5.78, 1.54–21.71), were significant.

Sensitivity analysis using Cox models showed that the results were about consistent with the primary analysis (Table S3 and S4).

## Discussion

In this study, we did not observe significant differences in all-cause mortality of PLHIV between the majority of initial ART regimens during the clinical care in China. After adjusting for potential confounders, only the differences between the EFV group and the NVP group, the DTG group and the NVP group, and the LPV group and the NVP group were found. After stratifying cause of mortality, we found that the differences in mortality between initial ART regimens were mainly observed in AIDS-related mortality.

In the multivariate Poisson regression analysis, we observed a higher rate of ART switch among patients who initiated NVP, DTG, or

**Table 1 | Characteristics of PLHIV included stratified by initial ART regimen**

| Variable | Overall | NNRTI | | | INSTI | | | PI | | p |
|---|---|---|---|---|---|---|---|---|---|---|
| | | EFV | NVP | RPV | DTG | RAL | EVG | LPV | DRV | |
| Total | 41018 | 35412 | 1303 | 123 | 1130 | 253 | 191 | 2592 | 14 | |
| Age (year, median [IQR]) | 37.00 [28.00, 52.00] | 38.00 [28.00, 52.00] | 47.00 [32.00, 59.00] | 31.00 [28.00, 37.50] | 34.00 [28.00, 44.00] | 35.00 [29.00, 46.00] | 34.00 [28.00, 47.00] | 35.00 [27.00, 47.00] | 34.00 [32.00, 46.00] | <0.001 |
| Age group (year, %) | | | | | | | | | | <0.001 |
| 18–24 | 5486 (13.4) | 4848 (13.7) | 102 (7.8) | 10 (8.1) | 129 (11.4) | 14 (5.5) | 20 (10.5) | 363 (14.0) | 0 (0.0) | |
| 25–34 | 12585 (30.7) | 10670 (30.1) | 285 (21.9) | 70 (56.9) | 471 (41.7) | 109 (43.1) | 76 (39.8) | 897 (34.6) | 7 (50.0) | |
| 35–49 | 11264 (27.5) | 9602 (27.1) | 355 (27.2) | 31 (25.2) | 341 (30.2) | 82 (32.4) | 54 (28.3) | 794 (30.6) | 5 (35.7) | |
| 50+ | 11683 (28.5) | 10292 (29.1) | 561 (43.1) | 12 (9.8) | 189 (16.7) | 48 (19.0) | 41 (21.5) | 538 (20.8) | 2 (14.3) | |
| Sex = Female (%) | 6309 (15.4) | 5342 (15.1) | 238 (18.3) | 6 (4.9) | 79 (7.0) | 33 (13.0) | 18 (9.4) | 593 (22.9) | 0 (0.0) | <0.001 |
| Route of transmission (%) | | | | | | | | | | <0.001 |
| Heterosexual | 17254 (42.1) | 15157 (42.8) | 774 (59.4) | 17 (13.8) | 295 (26.1) | 49 (19.4) | 28 (14.7) | 933 (36.0) | 1 (7.1) | |
| Homosexual | 17099 (41.7) | 14789 (41.8) | 446 (34.2) | 72 (58.5) | 576 (51.0) | 79 (31.2) | 99 (51.8) | 1030 (39.7) | 8 (57.1) | |
| Others | 6665 (16.2) | 5466 (15.4) | 83 (6.4) | 34 (27.6) | 259 (22.9) | 125 (49.4) | 64 (33.5) | 629 (24.3) | 5 (35.7) | |
| Region (%) | | | | | | | | | | <0.001 |
| Northern China | 6225 (15.2) | 4774 (13.5) | 23 (1.8) | 85 (69.1) | 354 (31.3) | 77 (30.4) | 88 (46.1) | 810 (31.2) | 14 (100.0) | |
| North-eastern China | 4099 (10.0) | 3770 (10.6) | 3 (0.2) | 1 (0.8) | 104 (9.2) | 1 (0.4) | 22 (11.5) | 198 (7.6) | 0 (0.0) | |
| Southern China | 5244 (12.8) | 4089 (11.5) | 44 (3.4) | 26 (21.1) | 316 (28.0) | 143 (56.5) | 44 (23.0) | 582 (22.5) | 0 (0.0) | |
| South-western China | 15706 (38.3) | 14332 (40.5) | 579 (44.4) | 3 (2.4) | 187 (16.5) | 6 (2.4) | 3 (1.6) | 596 (23.0) | 0 (0.0) | |
| Eastern China | 9744 (23.8) | 8447 (23.9) | 654 (50.2) | 8 (6.5) | 169 (15.0) | 26 (10.3) | 34 (17.8) | 406 (15.7) | 0 (0.0) | |
| Time to ART initiation (median days [IQR]) | 23.00 [11.00, 73.00] | 23.00 [11.00, 76.00] | 27.00 [12.00, 76.50] | 20.00 [12.00, 40.50] | 18.00 [9.00, 41.00] | 19.00 [11.00, 36.00] | 15.00 [8.00, 33.50] | 21.00 [11.00, 62.00] | 15.50 [9.00, 25.75] | <0.001 |
| Time to ART initiation (%) | | | | | | | | | | <0.001 |
| Same-day | 1385 (3.4) | 1196 (3.4) | 30 (2.3) | 4 (3.3) | 57 (5.0) | 8 (3.2) | 7 (3.7) | 83 (3.2) | 0 (0.0) | |
| 1–7 days | 5434 (13.2) | 4738 (13.4) | 147 (11.3) | 9 (7.3) | 172 (15.2) | 27 (10.7) | 32 (16.8) | 306 (11.8) | 3 (21.4) | |
| 8–30 days | 17264 (42.1) | 14684 (41.5) | 519 (39.8) | 71 (57.7) | 545 (48.2) | 143 (56.5) | 100 (52.4) | 1194 (46.1) | 8 (57.1) | |
| >30 days | 16935 (41.3) | 14794 (41.8) | 607 (46.6) | 39 (31.7) | 356 (31.5) | 75 (29.6) | 52 (27.2) | 1009 (38.9) | 3 (21.4) | |
| ART initiation year (%) | | | | | | | | | | <0.001 |
| 2017 | 17374 (42.4) | 15196 (42.9) | 699 (53.6) | 7 (5.7) | 212 (18.8) | 122 (48.2) | 3 (1.6) | 1135 (43.8) | 0 (0.0) | |
| 2018 | 15993 (39.0) | 14005 (39.5) | 444 (34.1) | 16 (13.0) | 420 (37.2) | 77 (30.4) | 56 (29.3) | 975 (37.6) | 0 (0.0) | |
| 2019 | 7651 (18.7) | 6211 (17.5) | 160 (12.3) | 100 (81.3) | 498 (44.1) | 54 (21.3) | 132 (69.1) | 482 (18.6) | 14 (100.0) | |
| CD4 at ART initiation (median [IQR]) | 254.00 [116.00, 372.00] | 261.00 [129.00, 377.00] | 183.00 [85.00, 264.00] | 348.00 [232.00, 456.00] | 159.00 [34.00, 327.00] | 74.00 [19.00, 233.00] | 257.00 [90.00, 400.00] | 219.00 [62.00, 365.00] | 67.34 [17.00, 199.00] | <0.001 |
| CD4 at ART initiation >500 (%) | 4622 (11.3) | 4116 (11.6) | 55 (4.2) | 22 (17.9) | 97 (8.6) | 11 (4.3) | 23 (12.0) | 297 (11.5) | 1 (7.1) | <0.001 |
| NRTI backbone (%) | | | | | | | | | | <0.001 |
| 3TC + TDF | 36140 (88.1) | 32439 (91.6) | 657 (50.4) | 75 (61.0) | 744 (65.8) | 207 (81.8) | 1 (0.5) | 2007 (77.4) | 10 (71.4) | |
| 3TC + AZT | 3828 (9.3) | 2768 (7.8) | 616 (47.3) | 4 (3.3) | 36 (3.2) | 16 (6.3) | 0 (0.0) | 386 (14.9) | 2 (14.3) | |
| Others | 1050 (2.6) | 205 (0.6) | 30 (2.3) | 44 (35.8) | 350 (31.0) | 30 (11.9) | 190 (99.5)[a] | 199 (7.7) | 2 (14.3) | |

*ART* antiretroviral therapy, *LTFU* loss to follow-up, *CD4 improvement* transition of people living with HIV from CD4 < 500 cells/μL to CD4 > 500 cells/μL, *EFV* efavirenz, *NVP* nevirapine, *RPV* rilpivirine, *DTG* dolutegravir, *RAL* raltegravie, *EVG* elvitegravir, *LPV* lopinavir, *DRV* darunavir, *NRTI* nucleotide reverse transcriptase inhibitor, *3TC* lamivudine, *TDF* tenofovir disproxil fumarate, *AZT* zidovudine.
[a]Emtricitabine (FTC)+ tenofovir alafenamide fumarate (TAF).

regimens based on other regimens compared to those who initiated EFV or LPV based, which contrasts with previous studies in high-income countries. A retrospective, observational study from a large-scale medical claims database in Japan[27] found that the switch rate constantly increased over four years for NNRTIs (17.8–45.2%) and PIs (16.2–47.6%) from 2011 to 2016, while INSTI maintained a low switch rate (2.3–7.6%). Similarly, a multicentre cohort study in the United States[28] found that compared to EFV/TDF/FTC users, atazanavir (ATV)/r + TDF/FTC users switched more (rate ratio [RR] = 1.80, 95% CI, 1.17–2.76), while those on DTG/ABC/3TC (RR [95% CI] = 0.16 [0.08–0.31]) or EVG/c/TAF/FTC (RR [95% CI] = 0.12 [0.06–0.27]) switched less. One possible reason for these findings is that long-term INSTI-based prescriptions are expensive in China. The cost of a bottle of DTG in China was CNY 1880, or approximately USD 274, which was a significant burden for patients (communication with infectious disease physicians). Therefore, once the condition of patients who started INSTI-based regimens was under control, more may tend to switch to the NFATP-sponsored regimens. Furthermore, the increased incidence of severe adverse events and risk of virological failure with those who initiated NVP-based ART, which has been extensively studied, may

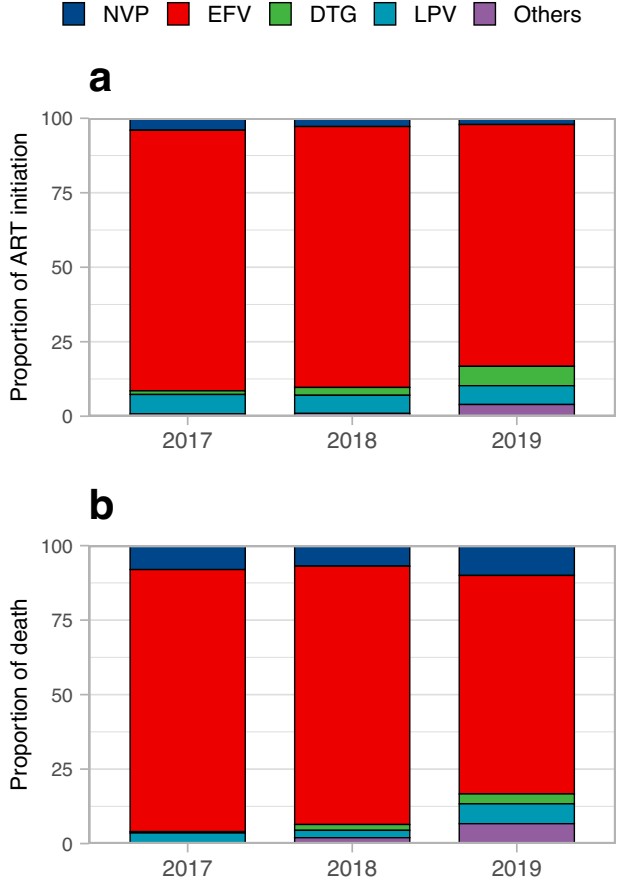

**Fig. 1 | Proportion of ART initiation and death among PLHIV in China 2017–2019.** Proportion of ART initiation (**a**) and death (**b**) among PLHIV in China 2017–2019, stratified by initial ART regimen, ART antiretroviral therapy, NVP nevirapine, EFV efavirenz, DTG dolutegravir, LPV lopinavir, Others= darunavir, raltegravie, elvitegravir and rilpivirine.

have contributed to the higher ART switch rate[29–35]. However, we acknowledge that alternative factors beyond those mentioned may contribute to the differences in ART switch rates. Additional research is needed to further explore and elucidate the underlying reasons for these discrepancies.

Consistent with previous studies, we found that patients who initiated EFV based ART regimen had lower LTFU rates. A large-scale cohort study in 62,500 adults with HIV in Europe and North America[19] found that LTFU was the lowest in those on EFV-based regimens, but otherwise broadly similar across the regimens. Another study, which analyzed data from 18 cohorts in Europe and North America from 2002–2009[36], found that compared to EFV, NVP and LPV were associated with higher rates of ART interruption for more than 1 month (subdistribution hazard ratios 1.5, 95% CI 1.3–1.7; and 1.4, 1.2–1.5, respectively). Following the WHO recommendations, EFV based ART regimen have been commonly used as first-line therapy for PLHIV in China since 2011 and were available through the NFATP[12,13]. The favorable efficacy and superior tolerability of this ART regimen in PLHIV in China were demonstrated in a cohort study carried by the China HIV/AIDS Clinical Trial Network (Clinicaltrials.gov ID: NCT01844297). It was reported that the high accessibility and efficacy were contributing factors to a low rate of LTFU[29–35].

We found that patients who initiated DTG, LPV and others-based ART regimens had significantly higher CD4 improvement rates compared to NVP based, which was similar to existing studies. A retrospective analysis in Thailand[37] found that compared with the EFV

group (patient who initiated EFV ART regimen), the DTG group showed greater increments of CD4 + (P < 0.001) T-cell counts at week 96. Another retrospective study of patients registering with the NFATP in Beijing, China from July 2012 to January 2017[38] found that patients in the LPV/ritonavir group were more likely to display improvements in CD4 + T-cell count over time than those in the EFV group (P < 0.001). The CD4 + T-cell count reflects the immune status and risk of opportunistic infections at the time of treatment for PLHIV. In addition, the favorable efficacy and safety profile of INSTIs compared with other drug classes were widely studied in randomised trials[3–5,39–45].

We found no significant differences in mortality between patients who initiated most of the ART regimens. A recent study in 62,500 adults with HIV in Europe and North America[19] found that there was little evidence that mortality rates differed between regimens with DTG, EVG, RPV, DRV, or EFV as the third drug. However, mortality was higher for RAL compared with DTG (aHR 1.49, 95% CI 1.15–1.94), EVG (1.86, 1.43–2.42), RPV (1.99, 1.49–2.66), DRV (1.62, 1.33–1.98), and EFV (2.12, 1.60–2.81) regimens[19]. Similar results were obtained in two other long-term observational studies[46,47]. Our finding was consistent with the previous literature in high-income countries, but as the data were from LMICs, this provided more evidence of the clinical outcomes of the initiation of INSTI-based ART. Notably, the differences in mortality were mainly observed in people who started NVP-based regimen in our study. As an ART drug recommended in 2009, NVP is currently predominantly used in LMICs[46,48]. Similar to EFV, another drug recommended in the same year, patients initiating ART with NVP tend to have a higher median age in our data due to the introduction of other new drugs over the past decade. Many research has reported the efficacy gap between NVP and other drugs[48,49], which is consistent with our findings. It is noteworthy that, compared to the other less common third drugs, the non-AIDS-related mortality among PLHIV initiating NVP and EFV has significantly decreased. This may be related to the side effects of less common drugs, warranting further research to elucidate the underlying causes of this phenomenon. Moreover, subsequent studies on the efficacy and mortality reduction of INSTIs, represented by DTG, will be very valuable, because China adopted DTG/3TC in the NFATP-sponsored regimens in 2021, which made it available free of charge for PLHIV from then on[50]. The government spending on HIV has steadily increased from 139 million USD in 2006 to 1,357 million USD in 2020[51]. However, given the heavy impact of the COVID-19 pandemic, which started in 2020 and spread globally, on the continuum of HIV care, including ART follow-up, the impact of the epidemic should be appropriately excluded in subsequent studies[52].

Our study was based on a large population between 2017 and 2019 covering a variety of regions in China, which ensured the representativeness of the sample and the robustness of the findings. To the best of our knowledge, this is the first cohort study to investigate the associations of modern initial first-line ART regimens with all-cause mortality in PLHIV in China. However, there are several limitations. First, it is important to note that our study is observational and does not demonstrate a causal relationship between initial ART regimen and clinical outcomes, as the treatment was not randomly assigned. There might have been some unmeasured or underlying causes that could bias our results. For example, given that all INSTIs and most PIs were not available free of charge before 2021 in China, patients' socioeconomic status could potentially affect our results. Due to the sensitivity of HIV data, obtaining and adjusting data on socioeconomic indicators such as income, employment status, and location of residence is challenging, which is consistent with similar studies[19,47]. If more data become available in the future, subsequent studies would further enhance our understanding of the topic. However, we adjusted for a range of demographic factors (particularly region) in our multivariable models using current available data, which we believe helps to mitigate the potential impact of socioeconomic status. Second, we were unable to consider the virological suppression or failure of PLHIV

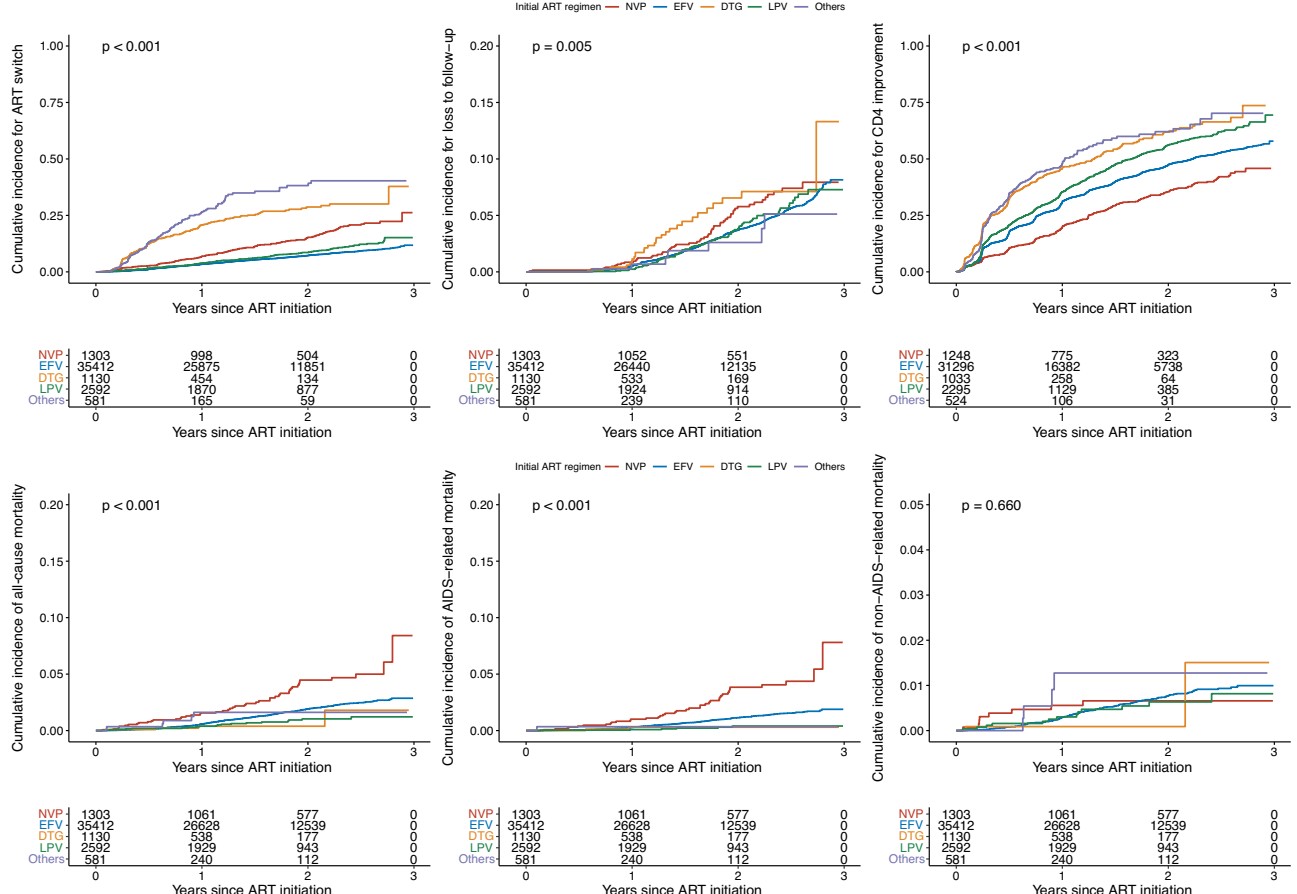

**Fig. 2 | Kaplan-Meier estimates of the cumulative incidence of six outcomes.** Kaplan-Meier estimates of the cumulative incidence of ART switch, LTFU, CD4 improvement, all-cause mortality, AIDS-related mortality, and non-AIDS related mortality. ART antiretroviral therapy, LTFU loss to follow-up, CD4 improvement= transition of people living with HIV from CD4 < 500 cells/µL to CD4 > 500 cells/µL, NVP nevirapine, EFV efavirenz, DTG dolutegravir, LPV lopinavir, Others= darunavir, raltegravie, elvitegravir and rilpivirine. Log-rank test, 2-sided no adjustment for multiple comparisons.

as a clinical outcome given that most of the viral load were missing in China. Third, the precision of our results was relatively low due to the small number of outcomes observed in certain groups (such as RPV, EVG and DRV groups). This was mainly due to the late introduction of these drugs in China, while the high price limited patients' consideration during our study period. Finally, our study period may be relatively short to fully capture the outcomes among PLHIV. However, given the implementation of the treat-all policy in China in 2016, and the significant impact of COVID-19 pandemic on the HIV care continuum in 2020[52,53], we had to limit the study duration to these three years. Additional long-term research is needed to further elucidate this topic after the lifting of zero-COVID policy in December 2022 in China.

In summary, significant differences in mortality rates of PLHIV between the majority of initial ART regimens were not observed in China. Our finding provided more evidence of the clinical outcomes of the initiation of INSTI-based ART. Further efforts are needed to confirm the reasons for the lack of significant improvement in mortality rates with the use of INSTIs in clinical practice.

## Methods

### Ethics approval

Ethics approval was obtained from the Ethics Review Committee for Biomedical Research, School of Public Health (Shenzhen), Sun Yat-sen University (Ref: 2020022). The data used in this study do not contain any identifying information available to the researchers.

### Study design and participants

A retrospective multicenter cohort study was performed using data extracted from the NFATP database. The database introduction could be found elsewhere[7]. Anonymised programmatic data on ART initiation and collection in PLHIV between January 1, 2017 and December 31, 2019 were collected from three provincial and municipal Centers for Disease Control and Prevention (CDCs) in Chongqing, Jiangsu and Dehong, and nine major infectious disease hospitals specialized in HIV care in Guangzhou, Shenzhen, Hohhot, Tianjin, Shenyang (2), Beijing, Shijiazhuang and Dalian in China. We included all subjects who were older than 18 years, started triple ART between January 1, 2017 and September 30, 2019, and had available baseline CD4 + T-cell count measurements (Fig. S1). The follow-up records were collected until study end (December 31, 2019).

Jiangsu is a province in Eastern China; Chongqing is a municipality and Dehong is an autonomous minority prefecture in Southwestern China. Guangzhou and Shenzhen are located in Southern China; Hohhot, Beijing, Tianjin and Shijiazhuang are cities in Northern China; Shenyang and Dalian are located in Northeastern China. These regions collectively represent 21.3% [41,018 / (57,194 + 64,170 + 71,204)] of newly reported HIV/AIDS cases in China during this period[54]. In China, CDCs at various levels are established to implement public health technical management and services, and are responsible for HIV surveillance. PLHIV are required to initiate ART and pick up drugs from designated infectious disease hospitals.

**Table 2 | Poisson regression model for the incidence of all-cause mortality, AIDS-related mortality, and non-AIDS-related mortality among people living with HIV on ART, adjusting for the main variables**

| | All-cause mortality | | AIDS-related mortality | | Non-AIDS-related mortality | |
|---|---|---|---|---|---|---|
| | aIRR (95% CI) | *p* value | aIRR (95% CI) | *p* value | aIRR (95% CI) | *p* value |
| **Age group (year)** | | | | | | |
| 18–24 | Ref. | | Ref. | | Ref. | |
| 25–34 | 1.42(1.08–1.88) | 0.012 | 3.12(1.89–5.15) | <0.001 | 0.92(0.64–1.34) | 0.679 |
| 35–49 | 3.04(2.37–3.90) | <0.001 | 5.22(3.15–8.38) | <0.001 | 1.65(1.21–2.26) | 0.002 |
| 50+ | 7.37(5.77–9.42) | <0.001 | 9.00(5.43–14.88) | <0.001 | 3.41(2.52–4.64) | <0.001 |
| **Sex** | | | | | | |
| Male | Ref. | | Ref. | | Ref. | |
| Female | 0.54(0.49–0.60) | <0.001 | 0.52(0.45–0.60) | <0.001 | 0.56(0.48–0.66) | <0.001 |
| **Route of transmission** | | | | | | |
| Heterosexual | Ref. | | Ref. | | Ref. | |
| Homosexual | 0.50(0.44–0.58) | <0.001 | 0.49(0.41–0.59) | <0.001 | 0.53(0.43–0.66) | <0.001 |
| Others | 0.71(0.57–0.87) | 0.001 | 0.38(0.27–0.55) | <0.001 | 1.15(0.88–1.51) | 0.314 |
| **Region** | | | | | | |
| Northern China | Ref. | | Ref. | | Ref. | |
| Northeastern China | 1.63(1.06–2.50) | 0.026 | 0.80(0.43–1.49) | 0.473 | 4.00(1.91–8.37) | <0.001 |
| Southern China | 0.22(0.11–0.45) | <0.001 | 0.11(0.01–0.76) | 0.025 | 0.31(0.11–0.89) | 0.029 |
| Southwestern China | 2.86(2.08–3.93) | <0.001 | 1.88(1.33–2.66) | <0.001 | 5.56(2.90–10.66) | <0.001 |
| Eastern China | 2.34(1.68–3.26) | <0.001 | 1.25(0.87–1.79) | 0.231 | 5.86(2.99–11.46) | <0.001 |
| **Time to ART initiation** | | | | | | |
| Same-day | Ref. | | Ref. | | Ref. | |
| 1–7 days | 1.22(0.90–1.65) | 0.199 | 0.90(0.63–1.28) | 0.548 | 2.05(1.16–3.64) | 0.014 |
| 8–30 days | 1.35(1.01–1.80) | 0.045 | 1.25(0.90–1.74) | 0.190 | 1.59(0.90–2.80) | 0.107 |
| >30 days | 2.00(1.50–2.67) | <0.001 | 1.64(1.18–2.29) | 0.003 | 3.04(1.73–5.32) | <0.001 |
| **ART year** | | | | | | |
| 2017 | Ref. | | Ref. | | Ref. | |
| 2018 | 0.86(0.80–0.93) | <0.001 | 0.82(0.75–0.91) | <0.001 | 0.88(0.78–0.99) | 0.035 |
| 2019 | 0.88(0.70–1.12) | 0.300 | 0.99(0.74–1.34) | 0.955 | 0.71(0.48–1.03) | 0.075 |
| **ART regimen** | | | | | | |
| NVP | Ref. | | Ref. | | Ref. | |
| EFV | 0.62(0.51–0.76) | <0.001 | 0.48(0.39–0.60) | <0.001 | 1.28(0.82–2.00) | 0.281 |
| DTG | 0.43(0.18–0.98) | 0.040 | 0.28(0.11–0.72) | 0.008 | 1.07(0.25–4.57) | 0.923 |
| LPV | 0.54(0.36–0.80) | 0.002 | 0.22(0.13–0.37) | <0.001 | 1.86(0.99–3.50) | 0.054 |
| Others | 1.44(0.57–3.64) | 0.442 | 0.34(0.05–2.38) | 0.279 | 6.53(2.05–20.80) | 0.002 |
| **CD4 at ART initiation (per 100)** | 0.67(0.64–0.69) | <0.001 | 0.69(0.67–0.72) | <0.001 | 0.61(0.58–0.65) | <0.001 |
| **NRTI backbone** | | | | | | |
| 3TC + TDF | Ref. | | Ref. | | Ref. | |
| 3TC + AZT | 0.77(0.62–0.96) | 0.020 | 1.01(0.78–1.32) | 0.913 | 0.47(0.30–0.72) | 0.001 |
| Others | 1.75(1.03–2.97) | 0.039 | 1.89(0.99–3.62) | 0.055 | 1.56(0.66–3.72) | 0.313 |

Poisson regression model with a time offset and robust variances was adjusted for main variables, including baseline age, sex, route of HIV acquisition, region, time to ART initiation, year of ART initiation, CD4 + T-cell counts, and NRTI backbone.
*ART* antiretroviral therapy, *NVP* nevirapine, *EFV* efavirenz, *DTG* dolutegravir, *LPV* lopinavir, *Others* darunavir, raltegravie, elvitegravir and rilpivirine, *aIRRs* adjusted incidence rate ratios, *NRTI* nucleotide reverse transcriptase inhibitor, *3TC* lamividine, *TDF* tenofovir disproxil fumarate, *AZT* zidovudine.

## Outcomes

Baseline and follow-up data, including social-demographics characteristics, clinical information, and laboratory test data, were extracted from the NFATP database. Details about the data were published elsewhere[10,55]. Baseline characteristics measured at ART initiation include age (18–24, 25–34, 35–49, 50+), sex (male and female, identified by the NFATP database based on the China ID card system), route of transmission (heterosexual, homosexual and others [including 90.4% intravenous drug use, 7.2% blood transfusion, 2.4% mother-to-child transmission]), region (Northern China [Hohhot, Shijiazhuang, Beijing and Tianjin], Northeastern China [Dalian and Shenyang], Southern China [Guangzhou and Shenzhen], Southwestern China

[Chongqing and Dehong], and Eastern China [Jiangsu]), time from diagnosis to ART initiation (time to ART initiation; same-day, 1–7 days, 8–30 days, >30 days), year of ART initiation (2017, 2018, 2019), and CD4 + T-cell counts. Follow-up data included initial ART regimen and NRTI backbone. Third ART drugs that were most commonly utilized in the study period included NVP, EFV, DTG, LPV, and others (including DRV, RAL, EVG and RPV). The NRTI backbone pairs were stratified as: 3TC and TDF, 3TC and AZT, and others.

The primary outcomes of interest were ART switch, retention, CD4 improvement, and mortality. ART switch refers to changes in third component for any reason that occurs during a patient's follow-up visit. Retention was represented by LTFU, which was defined by patients who

**Table 3 | Poisson regression model for the incidence of all-cause mortality, AIDS-related mortality, and non-AIDS-related mortality among people living with HIV on ART for each third drug comparison**

|  | All-cause mortality | | AIDS-related mortality | | Non-AIDS-related mortality | |
|---|---|---|---|---|---|---|
|  | aIRR (95% CI) | *p* value | aIRR (95% CI) | *p* value | aIRR (95% CI) | *p* value |
| EFV vs NVP | 0.64(0.53–0.79) | <0.001 | 0.49(0.40–0.62) | <0.001 | 1.35(0.86–2.11) | 0.198 |
| DTG vs NVP | 0.27(0.11–0.67) | 0.004 | 0.17(0.08–0.40) | <0.001 | 0.75(0.18–3.07) | 0.687 |
| LPV vs NVP | 0.49(0.33–0.75) | 0.001 | 0.20(0.12–0.33) | <0.001 | 1.68(0.83–3.43) | 0.151 |
| Others vs NVP | 2.22(0.51–9.74) | 0.291 | 0.29(0.08–1.07) | 0.062 | 7.74(1.03–58.17) | 0.047 |
| DTG vs EFV | 0.68(0.28–1.62) | 0.385 | 0.64(0.24–1.69) | 0.366 | 0.73(0.17–3.08) | 0.673 |
| LPV vs EFV | 0.88(0.62–1.27) | 0.499 | 0.46(0.29–0.74) | 0.001 | 1.48(0.90–2.42) | 0.119 |
| Others vs EFV | 2.39(0.88–6.48) | 0.088 | 0.71(0.11–4.80) | 0.726 | 5.78(1.54–21.71) | 0.009 |
| LPV vs DTG | 1.44(0.52–3.98) | 0.484 | 1.02(0.36–2.94) | 0.967 | 1.95(0.39–9.71) | 0.415 |
| Others vs DTG | 3.57(0.72–17.65) | 0.119 | 2.28(0.62–8.48) | 0.217 | 6.62(0.39–113.87) | 0.193 |
| Others vs LPV | 2.26(0.74–6.90) | 0.152 | 0.93(0.08–10.67) | 0.951 | 3.06(0.95–9.90) | 0.062 |

ART, antiretroviral therapy. NVP, nevirapine. EFV, efavirenz. DTG, dolutegravir. LPV, lopinavir. Others= darunavir, raltegravie, elvitegravir and rilpivirine. aIRR, adjusted incidence rate ratio. Poisson regression model with a time offset and robust variances was adjusted for main variables, including baseline age, sex, route of HIV acquisition, region, time to ART initiation, year of ART initiation, CD4 + T-cell counts, and NRTI backbone. NRTI, nucleotide reverse transcriptase inhibitor.

were >180 days (as recommended by a systematic review[56]) from the last clinical visit or drug pick-up during the study period. CD4 improvement was represented by changes in longitudinally-collected CD4 measurements. We defined CD4 improvement as transition of PLHIV from CD4 < 500 cells/μL to CD4 > 500 cells/μL. Mortality referred to all-cause mortality in PLHIV, including AIDS-related mortality and non-AIDS-related mortality.

### Statistical analyses

We used descriptive statistics to summarize baseline characteristics of PLHIV stratified by initial ART regimen. The median and IQR were used to describe continuous variables, and frequencies and proportions were used to describe categorical variables. Kruskal-Wallis tests were used to assess the associations of continuous variables with the initial ART class, and Pearson's chi-squared tests were used to assess the associations between categorical variables and the initial ART class.

We then used Kaplan-Meier curves to calculate stratified cumulative incidences of four outcomes, including ART switch, LTFU, CD4 improvement, and death. Individuals included in the study were censored at the time of settle in new areas, or when switched their first-line ART regimen, or when LTFU definition was met, or death, or when study end, whichever occurred first. Poisson regression model with a time offset and robust variances, adjusted for the main covariates, was used to estimate IRRs of outcomes by initial ART class[57,58]. To explore the association between initial ART regimen and different causes of death, we conducted the same analyses for both AIDS-related mortality and non-AIDS-related mortality. For all analyses, we fitted an unadjusted model and a model adjusted for the main variables, including baseline age, sex, route of HIV acquisition, region, time to ART initiation, year of ART initiation, CD4 + T-cell counts, and NRTI backbone.

To check the robustness of our findings, we conducted a sensitivity analysis using Cox models to reanalyse the data. All statistical tests were two-sided, and $P < 0.05$ was considered indicating strong evidence against the null hypothesis. All statistical analyses were conducted using R 4.2.1 (R Foundation for Statistical Computing, Vienna, Austria).

### Reporting summary

Further information on research design is available in the Nature Portfolio Reporting Summary linked to this article.

### Data availability

The raw data that support the findings of this study are not publicly available for confidentiality reasons, since these patients may be re-identified through various techniques, such as data linkage or combining datasets. The processed data are available on reasonable request to the corresponding author, H.Z., with each request subject to ethical and legislative review from the respective data sources. Source data are provided with this paper.

### Code availability

The R code used in the analyses was provided in 10.5281/zenodo.8240994. However, this study did not generate new or customized code/software. The Poisson regression models were fitted using the *glm* function from the *base* R package. The robust standard errors of Poisson models were computed using the *vcovHC* function from the *sandwich* R package. The Cox Proportional models were fitted using the *coxph* function from the *survival* R package.

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

## Acknowledgements

H.Z. is supported by the Shenzhen Science and Technology Innovation Commission Basic Research Program [JCYJ20190807155409373], the Natural Science Foundation of China Excellent Young Scientists Fund [82022064], Natural Science Foundation of China International/Regional Research Collaboration Project [72061137001], the Sanming Project of Medicine in Shenzhen [SZSM201811071], the High Level Project of Medicine in Longhua, Shenzhen [HLPM201907020105], Special Support Plan for High-Level Talents of Guangdong Province [2019TQ05Y230], the Fundamental Research Funds for the Central Universities [58000-31620005], Non-profit Central Research Institute Fund of Chinese Academy of Medical Sciences [2020-JKCS-030]. G.W. is supported by the Chongqing Talents Program for Innovative and Entrepreneurial Pioneers [cstc2021ycjh-bgzxm0097], the Chongqing Natural Science Foundation Project [cstc2021jcyj-msxmX1171], the Chinese State Key Laboratory of Infectious Disease Prevention and Control [2021SKLID303]. P.M. is supported by the Health Science and Technology Project of Tianjin Health Commission [ZC20037], the Tianjin Key Medical Discipline (Specialty) Construction Project [Infectious Diseases ZD02]. L.L. is supported by the National Key Research and Development Program of China [2022YFC2304800], the Science and Technology Project of Guangzhou [20220020285]. X.H. is supported by the Public Health Talent Grant by Beijing Municipal Health Commission [Global Health Governance-02-12], the Capital Health Development Research [2022-2-2185; 2022-1G-3011]. B.S. is supported by the High-Level Public Health Specialized Talents Project of Beijing Municipal Health Commission [2022-2-018], the National Key R&D Program of China [2021YFC2301900; 2021YFC2301905], the Beijing Key Laboratory for HIV/AIDS Research [BZ0089]. T.Z. is supported by the High-Level Public Health Specialized Talents Project of Beijing Municipal Health Commission [2022-1-007], the Climbing the Peak (Dengfeng) Talent Training Program of Beijing Hospitals Authority (DFL20191701). All funding parties did not have any role in the design of the study or in the explanation of the data. We thank the Prof. Junfeng Wang from Utrecht University, Netherlands for supporting the statistical methods and R codes in our study. We thank Yi-Fan Lin from the School of Public Health (Shenzhen), Sun Yat-sen University for supporting the statistical methods in our study.

## Author contributions

X.W. conceived and designed the study in consultation with H.Z., H.Z., G.W., P.M., R.W., L.L., J.X., Q.L., Y.Y., L.W., X.X., Y.Q., G.F., X.H., B.S., T.Z., and H.W. contributed to data collection. X.W., Y.C., and Y.L. contributed to data analysis and presentation. X.W. drafted the report with all authors critically reviewing the paper. All authors saw and approved the final report.

## Competing interests

The authors declare no competing interests.

## Additional information

[1]Shenzhen Campus of Sun Yat-sen University, Shenzhen, PR China. [2]School of Public Health (Shenzhen), Sun Yat-sen University, Shenzhen, PR China. [3]Institute for AIDS/STD Control and Prevention, Chongqing Center for Disease Control and Prevention, Chongqing, PR China. [4]Department of Infectious Diseases, Tianjin Second People's Hospital, Tianjin, PR China. [5]Tianjin Association of STD/AIDS Prevention and Control, Tianjin, PR China. [6]Dalian Public Health Clinical Center, Dalian, PR China. [7]Infectious Disease Center, Guangzhou Eighth People's Hospital, Guangzhou Medical University, Guangzhou, PR China. [8]Clinical Research Academy, Peking University Shenzhen Hospital, Peking University, Shenzhen, PR China. [9]Dehong Prefecture Center for Disease Control and Prevention, Dehong, PR China. [10]Shijiazhuang Fifth Hospital, Shijiazhuang, PR China. [11]No.6 People's Hospital of Shenyang, Shenyang, PR China. [12]No.2 Hospital of Hohhot, Hohhot, PR China. [13]Department of STD/AIDS Control and Prevention, Jiangsu Provincial Center for Disease Control and Prevention, Nanjing, PR China. [14]Clinical and Research Center for Infectious Diseases, Beijing Youan Hospital, Capital Medical University, Beijing, PR China. [15]National Clinical Research Centre for Infectious Diseases, The Third People's Hospital of Shenzhen and The Second Affiliated Hospital of Southern University of Science and Technology, Shenzhen, PR China. [16]School of Public Health, Fudan University, Shanghai, PR China. [17]School of Public Health, Southwest Medical University, Luzhou, PR China. [18]Kirby Institute, University of New South Wales, Sydney, NSW, Australia. [19]These authors contributed equally: Xinsheng Wu, Guohui Wu, Ping Ma, Rugang Wang, Linghua Li. ✉e-mail: fugf@jscdc.cn; huangxiaojie78@ccmu.edu.cn; binsu@ccmu.edu.cn; zt_doc@ccmu.edu.cn; huiwang98@szsy.sustech.edu.cn; hzou@kirby.unsw.edu.au

