## [Peer Review File · Nature Communications]

Associations of modern initial antiretroviral therapy regimens with all-cause mortality in people living with HIV in resource-limited settings: a retrospective multicenter cohort study in ChinaReviewers' Comments:

Reviewer #1:

Remarks to the Author:

The manuscript by Wu et al. "Associations of modern initial antiretroviral therapy 2 regimens with all-cause mortality in people living with 3 HIV in resource-limited settings: a retrospective 4 multicenter cohort study in China" reported a large, multi-region retrospective study comparing all-cause mortality among PLHIV in China who were on different initial ART regimens between 2017 and 2019. The paper was well written and raised interesting issues, with some concerns needed to be addressed.

1. Should provide a summarized conclusion in the "Abstract".
2. Given that all INSTIs and most PIs were not available free of charge before 2021 in China, patients' social economic status or its proxy could be a critical confounding variable to be adjusted for when comparing the impact of NVP-based regimen versus LPV, DTG and other drugs-based regimens on CD4 improvement and mortality. This should be discussed.
3. The reason per se, raised by the authors to explain a higher rate of ART switch among patients who initiated NVP, DTG, or regimens based on other regimens compared to those who initiated EFV or LPV based is controversial. (Line 211-216, page 9)
4. The study period is relatively short for investigating the associations of modern initial first-line ART regimens with all-cause mortality (the main study objective) in PLWH in China, as AIDS-related deaths have been dramatically decreasing due to the overall effectiveness of treat-all policy (no matter what the initiated first line regimen is) and aging-related non-communicable chronic disease have been increasing taking over as the leading causes of death. This should be discussed as a limitation of the study.
5. The major outcome of the study was all-cause mortality. However, given the differential efficacy and side effects of different ARVs, it is worthy investigating the associations of ART regimens with cause-specific mortality.

Reviewer #2:

Remarks to the Author:

This paper comparing outcomes for PLHIV starting on different ART regimens in China is robust.

I realise that English is unlikely to be the first-language of the authors, so I would advise the journal staff to do some English-language editing. However, the meaning of the author's sentences are still clear.

I would advise the authors to check through the paper for sentences that are unreferenced. For example:

"Nonnucleoside reverse-transcriptase (NNRTI)- and PI-based ART regimens were used the most commonly in LMICs."

"These regions collectively represent 20.8% of newly reported HIV/AIDS cases in China in 2019."

The authors state: "All patients were followed-up until study end (December 31, 2019)."

This is not correct as you later on say that some individuals were censored before this.

You also later say "A total of 41,018 individuals were included in the analyses from January 2017 to September 2019." I believe you mean to say individuals that started ART between January 2017 and September 2019.

"To explore the possibility of reverse causality, we stratified the follow-up time into a short-term stratum of ≤ 2 years and a long-term stratum of > 2 years."

This is confusing. How could subsequent mortality cause someone to have been on a particular

regimen beforehand? Do you mean that you are examining preferential prescribing? This analysis is not well explained and doesn't seem necessary. It also seems under-powered. I would remove it.

"For all analyses, we fitted an unadjusted model and a model adjusted for the main variables, including baseline age, gender, route of HIV acquisition, region, time to ART initiation, year of ART initiation, year after ART initiation, CD4+ T-cell counts, and NRTI backbone."
Why would you include the "Year after ART initiation"

Why did you use Poisson regression rather than Cox regression?

You should report in the results how many people were loss to follow-up, how many switched regimen, and how many achieved a CD4>500.

"P<0.05 was considered statistically significant."

Please try to avoid using the term "statistically significant. See here for more information:

<https://www.nature.com/articles/d41586-019-00857-9>

<https://www.nature.com/articles/d41586-019-00874-8>

<https://www.bmj.com/content/322/7280/226.1>

In the analysis of "Cumulative incidence for CD4 improvement" - are all people included? If someone starts ART with a CD4>500 then they should be excluded from this analysis.

Table 1: What are the units for "Time to ART initiation"? Time to ART (%) is presumably also time to ART initiation.

99.5% of people on EVG are receiving an "other" backbone. What backbone is this?

Supplementary table 2 needs to be reformatted so the column indicating what the variables are is on the same page as the columns for CD4 improvement and mortality. I would move this table to the main paper.

"The proportion of PLHIV deaths and the mortality rate (per 1000 person-years) from 2017 to 2019 were shown in Figure 1B."

The mortality rate is not shown in figure 1B. You should the proportion of PLHIV deaths by initial ART regimen.

In the sentence starting "In the multivariate Poisson regression model" you should define the comparator groups for the IRRs.

"We found no significant differences between patients who initiated most of the ART regimens."

You need to clarify you mean differences in mortality. You found a difference for NVP. Why do you think there was this difference? Your table 1 shows that people on NVP were much older than on other regimens. This is not mentioned in the article. Why are people starting ART on NVP so much older than for other regimens?

People who acquired HIV through injecting drug use were grouped into the "other" acquisition route category. However, they often have much worse prognosis. How many people included in the study acquired HIV through IDU?

In the discussion you raise these important points:

"One possible reason for these findings is that long-term INSTI-based prescriptions are expensive in China."

"However, the landscape of fully reimbursed ART in China before 2021 included two NNRTIs (EFV and

NVP), and a single PI (LPV); along with several NRTIs. All INSTIs and most PIs were not available free of charge before 2021"

"Therefore, INSTIs, including DTG, were mainly used for PLHIV who are more severely ill at the time of initiating ART and usually with CD4<200 cells/ μ L." (reference needed)

I think it would be helpful to include this information into the introduction to help the reader better understand the context.

Was there any missing data? If so, how was this dealt with?

Response to the reviewers:

Reviewer #1 (Remarks to the Author):

The manuscript by Wu et al. “Associations of modern initial antiretroviral therapy 2 regimens with all-cause mortality in people living with 3 HIV in resource-limited settings: a retrospective 4 multicenter cohort study in China” reported a large, multi-region retrospective study comparing all-cause mortality among PLHIV in China who were on different initial ART regimens between 2017 and 2019. The paper was well written and raised interesting issues, with some concerns needed to be addressed.

Authors’ reply:

Thank you for this positive feedback.

1. Should provide a summarized conclusion in the “Abstract”.

Authors’ reply:

According to the reviewer’s suggestion, we have added a summarized conclusion, and revised other parts to meet the word limit requirements of the Abstract section (page 4; line 77-90):

“Despite the proven virological advantages, there remains some controversy regarding whether first-line integrase strand transfer inhibitors (INSTIs)-based antiretroviral therapy (ART) contributes to reducing mortality of people living with HIV (PLHIV) in clinical practice. Here we report a retrospective study comparing all-cause mortality among PLHIV in China who were on different initial ART regimens (nevirapine, efavirenz, dolutegravir, lopinavir, and others [including darunavir, raltegravir, elvitegravir and rilpivirine]) between 2017 and 2019. A total of 41,018 individuals were included across China, representing 21.3% of newly reported HIV/AIDS cases collectively in the country during this period. Only the differences in all-cause mortality of PLHIV between the

efavirenz group and the nevirapine group, the dolutegravir group and the nevirapine group, and the lopinavir group and the nevirapine group, were observed in China. After stratifying the cause of mortality, we found that the differences in mortality between initial ART regimens were mainly observed in AIDS-related mortality.”

2. Given that all INSTIs and most PIs were not available free of charge before 2021 in China, patients’ social economic status or its proxy could be a critical confounding variable to be adjusted for when comparing the impact of NVP-based regimen versus LPV, DTG and other drugs-based regimens on CD4 improvement and mortality. This should be discussed.

Authors’ reply:

We fully agree with the reviewer that the socioeconomic status could potentially affect our results and should be discussed. Due to the sensitivity of HIV data, in line with similar studies [1, 2], obtaining and adjusting data on socioeconomic indicators such as income, employment status, and location of residence is challenging. However, we adjusted for a range of demographic factors (particularly region) in our multivariable models using current available data, which we believe helps to mitigate the potential impact of socioeconomic status.

We have updated our Discussion section to acknowledge the limitations regarding the potential impact of unmeasured factors (particularly socioeconomic status) on our results, and emphasized that if more data become available in the future, subsequent studies would further enhance our understanding of the topic (page 12; line 308-316):

“For example, given that all INSTIs and most PIs were not available free of charge before 2021 in China, patients’ socioeconomic status could potentially affect our results. Due to the sensitivity of HIV data, obtaining and adjusting data on socioeconomic indicators such as income, employment status, and location of residence is challenging, which is consistent with similar studies^{19,53}. If more data become available in the future, subsequent studies

would further enhance our understanding of the topic. However, we adjusted for a range of demographic factors (particularly region) in our multivariable models using current available data, which we believe helps to mitigate the potential impact of socioeconomic status.”

3. The reason per se, raised by the authors to explain a higher rate of ART switch among patients who initiated NVP, DTG, or regimens based on other regimens compared to those who initiated EFV or LPV based is controversial. (Line 211-216, page 9)

Authors' reply:

We thank the reviewer for bringing up this important issue. We have revised the Discussion section of our manuscript to acknowledge that alternative factors beyond those mentioned may contribute to this phenomenon and warranted further investigation (page 10; line 242-245):

“However, we acknowledge that alternative factors beyond those mentioned may contribute to the differences in ART switch rates. Additional research is needed to further explore and elucidate the underlying reasons for these discrepancies.”

4. The study period is relatively short for investigating the associations of modern initial first-line ART regimens with all-cause mortality (the main study objective) in PLWH in China, as AIDS-related deaths have been dramatically decreasing due to the overall effectiveness of treatment policy (no matter what the initiated first line regimen is) and aging-related non-communicable chronic disease have been increasing taking over as the leading causes of death. This should be discussed as a limitation of the study.

Authors' reply:

We fully agree that the shorter study period may limit the ability to fully capture the impact of ART regimens on mortality. However, to avoid population-level interventions that could have

affected the results of our study, we had to limit the study duration to three years (2017-2019). Specifically, the implementation of the treat-all policy in China in 2016 adjusted the treatment eligibility for PLHIV [3], and the COVID-19 pandemic in 2020 had significant impacts on the HIV care continuum [4, 5]. We have included a discussion of this limitation in the revised manuscript and emphasized the need for long-term studies after the lifting of zero-COVID policy in December 2022 in China (page 13; line 322-327):

“Finally, our study period may be relatively short to fully capture the outcomes among PLHIV. However, given the implementation of the treat-all policy in China in 2016, and the significant impact of COVID-19 on the HIV care continuum in 2020^{52,54}, we had to limit the study duration to these three years. Additional long-term research is needed to further elucidate this topic after the lifting of zero-COVID policy in December 2022 in China.”

We also acknowledge the increasing prevalence of non-communicable chronic diseases among PLHIV, and have included an analysis of cause-specific mortality as suggested by the reviewer:

Figure 2 (part). Kaplan-Meier estimates of the cumulative incidence of ART switch, LTFU, CD4 improvement, all-cause mortality, AIDS-related mortality, and non-AIDS related mortality

Table 3 (part). Poisson regression model for the incidence of all-cause mortality, AIDS-related mortality, and non-AIDS-related mortality among people living with HIV on ART for each third drug comparison

	AIDS-related mortality		Non-AIDS-related mortality	
	aIRR (95% CI)	p-value	aIRR (95% CI)	p-value
EFV vs NVP	0.49(0.40-0.62)	<0.001	1.35(0.86-2.11)	0.198
DTG vs NVP	0.17(0.08-0.40)	<0.001	0.75(0.18-3.07)	0.687
LPV vs NVP	0.20(0.12-0.33)	<0.001	1.68(0.83-3.43)	0.151
Others vs NVP	0.29(0.08-1.07)	0.062	7.74(1.03-58.17)	0.047
DTG vs EFV	0.64(0.24-1.69)	0.366	0.73(0.17-3.08)	0.673
LPV vs EFV	0.46(0.29-0.74)	0.001	1.48(0.90-2.42)	0.119
Others vs EFV	0.71(0.11-4.80)	0.726	5.78(1.54-21.71)	0.009
LPV vs DTG	1.02(0.36-2.94)	0.967	1.95(0.39-9.71)	0.415
Others vs DTG	2.28(0.62-8.48)	0.217	6.62(0.39-113.87)	0.193
Others vs LPV	0.93(0.08-10.67)	0.951	3.06(0.95-9.90)	0.062

ART, antiretroviral therapy. NVP, nevirapine. EFV, efavirenz. DTG, dolutegravir. LPV, lopinavir. Others= darunavir, raltegravir, elvitegravir and rilpivirine. aIRR, adjusted incidence rate ratio. Poisson regression model with a time offset and robust variances was adjusted for main variables, including baseline age, sex, route of HIV acquisition, region, time to ART initiation, year of ART initiation, CD4+ T-cell counts, and NRTI backbone. NRTI, nucleotide reverse transcriptase inhibitor.

We appreciate the insightful suggestions from the reviewer and find that this updated result provides innovative insights into the effectiveness of different ART regimens in real-world practice.

5. The major outcome of the study was all-cause mortality. However, given the differential efficacy and side effects of different ARVs, it is worthy investigating the associations of ART regimens with cause-specific mortality.

Authors' reply:

Thank you for your constructive feedback on our manuscript. We have conducted additional analyses on cause-specific mortality (AIDS-related mortality and non-AIDS-related mortality),

and have included these results in the revised manuscript. After stratifying mortality, we found that the differences in mortality between initial ART regimens were mainly observed in AIDS-related mortality. We find that this suggestion is highly valuable and has helped to improve the innovation and interpretability of our study results.

Reviewer #2 (Remarks to the Author):

This paper comparing outcomes for PLHIV starting on different ART regimens in China is robust.

Authors' reply:

Thank you for this positive feedback.

I realise that English is unlikely to be the first-language of the authors, so I would advise the journal staff to do some English-language editing. However, the meaning of the author's sentences are still clear.

I would advise the authors to check through the paper for sentences that are unreferenced. For example:

"Nonnucleoside reverse-transcriptase (NNRTI)- and PI-based ART regimens were used the most commonly in LMICs."

"These regions collectively represent 20.8% of newly reported HIV/AIDS cases in China in 2019."

Authors' reply:

Thank you for your comment. We have carefully reviewed our manuscript and added appropriate references to the examples you mentioned (page 6; line 127-129) (page 14; line 352-354), as well as other similar sentences (page 5; line 102-104).

The authors state: "All patients were followed-up until study end (December 31, 2019)."

This is not correct as you later on say that some individuals were censored before this.

You also later say "A total of 41,018 individuals were included in the analyses from January 2017 to September 2019." I believe you mean to say individuals that started ART between January 2017 and September 2019.

Authors' reply:

Thank you for your comment. We have revised our manuscript accordingly to ensure clarity (page 14; line 346-347) (page 6; line 144-145):

"The follow-up records were collected until study end (December 31, 2019)."

"A total of 41,018 individuals who started ART between January 2017 and September 2019 were included in the analyses."

"To explore the possibility of reverse causality, we stratified the follow-up time into a short-term stratum of ≤ 2 years and a long-term stratum of > 2 years."

This is confusing. How could subsequent mortality cause someone to have been on a particular regimen beforehand? Do you mean that you are examining preferential prescribing? This analysis is not well explained and doesn't seem necessary. It also seems under-powered. I would remove it.

Authors' reply:

Thank you for your constructive feedback on our manuscript. To clarify, our aim was to explore the possibility that a frailty bias could have led to preferential prescribing of certain regimens to patients who were at higher risk of death in the short term. After careful consideration and further examination of our data, we have come to the conclusion that this analysis was not necessary for our study and have removed this analysis from our manuscript.

"For all analyses, we fitted an unadjusted model and a model adjusted for the main variables, including baseline age, gender, route of HIV acquisition, region, time to ART initiation, year of ART initiation, year after ART initiation, CD4+ T-cell counts, and NRTI backbone."

Why would you include the "Year after ART initiation"

Authors' reply:

Thank you for bringing this to our attention. We have corrected this error in the revised manuscript.

Why did you use Poisson regression rather than Cox regression?

Authors' reply:

Thank you for your question. It is important to note that Cox regression models assumes that the hazard ratio (HR) is constant over time. We can see clearly cross of our KM curves, which indicates that the assumption is violated for our data. Moreover, we conducted a Schoenfeld test and found that $p < 0.05$, suggesting violations of the assumption (see figure below). In the presence of time-dependent covariates (baseline age, route of transmission, etc.) or non-proportional hazards, the hazard ratio is no longer constant over time, and thus the results of the Cox regression may be biased.

In contrast, Poisson regression with a time offset and robust variances, a widely recognized adjustment [6, 7] that has been used in many high-impact studies [8, 9], can relax the assumption that the outcome follows a Poisson distribution and directly estimate IRRs from binary endpoints. Therefore, we use Poisson regression rather than Cox regression. This methodological choice aligns with the practice in the field of HIV research [10] and ensures the rigor of our analysis. We have added references that support the adjustment approach in the revised manuscript.

To further address your concerns, we also added a sensitivity analysis using Cox models to check the robustness of our findings in the revised manuscript, and found that the results were about consistent with those obtained from the Poisson regression models (see revised Supplementary Tables S3 and S4).

You should report in the results how many people were loss to follow-up, how many switched regimen, and how many achieved a CD4>500.

Authors' reply:

Thank you for the comment. We have added them in the Results section in the revised manuscript (page 7; line 148-152):

“2,777 (6.8%) switched regimen, 1,132 (2.8%) loss to follow-up (LTFU), 14,366 (39.5%) achieved a CD4>500 from the 36,396 patients with initial CD4<500, and 656 (1.6%; 393 AIDS-related death and 263 non-AIDS-related death) died during 65,578 person-years of follow-up (median 1.6 years [IQR 1.0-2.2]).”

"P<0.05 was considered statistically significant."

Please try to avoid using the term "statistically significant. See here for more information:

<https://www.nature.com/articles/d41586-019-00857-9>

<https://www.nature.com/articles/d41586-019-00874-8>

<https://www.bmj.com/content/322/7280/226.1>

Authors' reply:

Thank you for the reminder. We have revised the statement to “P < 0.05 was considered indicating strong evidence against the null hypothesis” (page 16; line 408-409), and avoided this term in the manuscript.

In the analysis of "Cumulative incidence for CD4 improvement" - are all people included? If someone starts ART with a CD4>500 then they should be excluded from this analysis.

Authors' reply:

In the analysis of cumulative incidence for CD4 improvement, as shown in the Figure 2, only

36,396 individuals with baseline CD4<500 were included. We have added this explanation to the Results section in the revised manuscript (page 7; line 149-153):

“...14,366 (39.5%) achieved a CD4>500 from the 36,396 patients with initial CD4<500...”

“In the analysis of cumulative incidence for CD4 improvement, only 36,396 individuals with initial CD4<500 were included.”

Table 1: What are the units for "Time to ART initiation"? Time to ART (%) is presumably also time to ART initiation.

Authors' reply:

The units for “time to ART initiation” are days. We have added it to the Table 1, and revised “Time to ART (%)” to “Time to ART initiation (%)” in the revised tables.

99.5% of people on EVG are receiving an "other" backbone. What backbone is this?

Authors' reply:

Thank you for the reminder. It is emtricitabine (FTC)+ tenofovir alafenamide fumarate (TAF). We have added the clarification to the footnote in the revised Table 1.

Supplementary table 2 needs to be reformatted so the column indicating what the variables are is on the same page as the columns for CD4 improvement and mortality. I would move this table to the main paper.

Authors' reply:

Thank you for your suggestion. We have reformatted Table S2 accordingly. Considering the

analyses for causes of mortality we have added, and the aim of our study, we have decided to move the results for all-cause mortality, AIDS-related mortality, and non-AIDS-related mortality from the Table S2 to the main paper as Table 2.

"The proportion of PLHIV deaths and the mortality rate (per 1000 person-years) from 2017 to 2019 were shown in Figure 1B."

The mortality rate is not shown in figure 1B. You should the proportion of PLHIV deaths by initial ART regimen.

Authors' reply:

Thank you for the reminder. We have corrected this error in the revised manuscript.

In the sentence starting "In the multivariate Poisson regression model" you should define the comparator groups for the IRRs.

Authors' reply:

Thank you for the reminder. We have defined the comparator groups in this sentence in the revised manuscript (page 8; line 181-189):

"In the multivariate Poisson regression model, age at 35-49 years (adjusted incidence rate ratio [aIRR] 3.04, 2.37-3.90; Reference: 18-24 years; Table 2 and Figure 2), 50+ years (7.37, 5.77-9.42), location in the Southwestern (2.86, 2.08-3.93; Reference: Northern China) and Eastern China (2.34, 1.68-3.26), and >30 days to ART initiation (2.00, 1.50-2.67; Reference: same-day initiation) were risk factors for increased incidence of all-cause mortality; whereas female (0.54, 0.49-0.60; Reference: male), homosexual transmission (0.50, 0.44-0.58; Reference: heterosexual), location in the Southern China (0.22, 0.11-0.45; Reference: Northern China), and initial ART in 2018 (0.86, 0.80-0.93; Reference: 2017)

were protective factors.”

"We found no significant differences between patients who initiated most of the ART regimens."

You need to clarify you mean differences in mortality. You found a difference for NVP. Why do you think there was this difference? Your table 1 shows that people on NVP were much older than on other regimens. This is not mentioned in the article. Why are people starting ART on NVP so much older than for other regimens?

Authors' reply:

Thank you for the reminder. We have revised it accordingly to clarify it is “differences in mortality”. Regarding the difference in mortality for NVP, and the difference in age for people who initiated NVP, we have added discussions in the revised manuscript (page 11; line 284-290):

“Notably, the differences in mortality were mainly observed in people who started NVP-based regimen in our study. As an ART drug recommended in 2009, NVP is currently predominantly used in LMICs^{46,48}. Similar to EFV, another drug recommended in the same year, patients initiating ART with NVP tend to have a higher median age in our data due to the introduction of other new drugs over the past decade. Many research has reported the efficacy gap between NVP and other drugs^{48,49}, which is consistent with our findings.”

People who acquired HIV through injecting drug use were grouped into the "other" acquisition route category. However, they often have much worse prognosis. How many people included in the study acquired HIV through IDU?

Authors' reply:

We thank the reviewer for bringing up this important issue. In “other” category, there were 90.4% individuals who acquired HIV through IDU, 7.2% thorough blood transfusion, and 2.4% through mother-to-child transmission. We have added it to the Method section in the revised manuscript (page 14; line 364-365).

In the discussion you raise these important points:

"One possible reason for these findings is that long-term INSTI-based prescriptions are expensive in China."

"However, the landscape of fully reimbursed ART in China before 2021 included two NNRTIs (EFV and NVP), and a single PI (LPV); along with several NRTIs. All INSTIs and most PIs were not available free of charge before 2021"

"Therefore, INSTIs, including DTG, were mainly used for PLHIV who are more severely ill at the time of initiating ART and usually with CD4<200 cells/ μ L." (reference needed)

I think it would be helpful to include this information into the introduction to help the reader better understand the context.

Authors' reply:

Thank you for the insightful suggestion. We have included this information into the Introduction section and added a reference in the revised manuscript (page 6; line 131-136). The statement “Therefore, INSTIs, including DTG, were mainly used for PLHIV who are more severely ill at the time of initiating ART and usually with CD4<200 cells/ μ L.” was derived from communications we had with infectious disease physicians. We realized that there is no specific reference to support this statement and have decided to remove it from the manuscript.

Was there any missing data? If so, how was this dealt with?

Authors' reply:

Our study is based on a well-established national database that has been diligently managed and maintained since 2002 [11]. As a result, the data we obtained for all variables, except for baseline CD4 counts (5.3% missing) and other laboratory variables such as HIV viral load (82.1% missing), liver function (42.5% missing), etc., did not have any missing values.

For the baseline CD4 count, we employed a matching procedure by pairing the missing CD4 count results with the closest measurement within a three-day window of the baseline assessment. After that, there were 491 patients who did not have a match within the designated three-day window, subsequently resulting in their exclusion from the analysis. We have added a patient flowchart (Figure S1) in the revised version to further promote transparency of our data processing:

We have also added a statement in the Methods section about the exclusion criteria regarding

baseline CD4 count (page 14; line 344-346):

“We included all subjects who...and had available baseline CD4+ T-cell count measurements.”

For other laboratory data, we decided not to include these variables in our analysis in order to uphold the integrity and reliability of the data.

References

1. Trickey, A., et al., *Associations of modern initial antiretroviral drug regimens with all-cause mortality in adults with HIV in Europe and North America: a cohort study*. The Lancet. HIV, 2022. **9**(6): p. e404-e413.
2. Cole, S.R., et al., *Incident AIDS or Death After Initiation of Human Immunodeficiency Virus Treatment Regimens Including Raltegravir or Efavirenz Among Adults in the United States*. Clinical Infectious Diseases : an Official Publication of the Infectious Diseases Society of America, 2017. **64**(11): p. 1591-1596.
3. Chen, Q., et al., *Effect of Late Testing and Antiretroviral Treatment on Mortality Among People Living With HIV in the Era of Treat-All in Guangdong Province, China, 1992-2018: A Cohort Study*. Frontiers In Public Health, 2022. **10**: p. 851117.
4. Wu, X., et al., *The impact of COVID-19 non-pharmaceutical interventions on HIV care continuum in China: An interrupted time series analysis*. The Lancet Regional Health. Western Pacific, 2022. **29**: p. 100569.
5. Wu, X., et al., *The Impact of COVID-19 Lockdown on Cases of and Deaths From AIDS, Gonorrhoea, Syphilis, Hepatitis B, and Hepatitis C: Interrupted Time Series Analysis*. JMIR Public Health and Surveillance, 2023. **9**: p. e40591.
6. Zou, G., *A modified poisson regression approach to prospective studies with binary data*. American Journal of Epidemiology, 2004. **159**(7): p. 702-706.
7. Zou, G.Y. and A. Donner, *Extension of the modified Poisson regression model to prospective studies with correlated binary data*. Statistical Methods In Medical Research, 2013. **22**(6): p. 661-670.
8. Richterman, A., et al., *The effects of cash transfers on adult and child mortality in low- and middle-income countries*. Nature, 2023. **618**(7965): p. 575-582.
9. Chihara, D., et al., *Early drug development in solid tumours: analysis of National Cancer Institute-sponsored phase 1 trials*. Lancet (London, England), 2022. **400**(10351): p. 512-521.
10. Mody, A., et al., *Longitudinal engagement trajectories and risk of death among new*

ART starters in Zambia: A group-based multi-trajectory analysis. PLoS Medicine, 2019. **16**(10): p. e1002959.

11. Ma, Y., et al., *Cohort profile: the Chinese national free antiretroviral treatment cohort.* International Journal of Epidemiology, 2010. **39**(4): p. 973-979.

Reviewers' Comments:

Reviewer #1:

Remarks to the Author:

Most of my concerns have been satisfactorily addressed or acknowledged in the revised manuscript. As shown in the Author's Responses, the additional cause-specific analyses comparing other drugs with NVP and EFV actually showed significant benefits of NVP and EFV on non-AIDS-related mortality in the real-world setting. Should this be acknowledged or mentioned, the paper is acceptable.

Reviewer #2:

Remarks to the Author:

The authors have adequately responded to all of my previous comments. I have no further comments.

Response to the reviewers:

Reviewer #1 (Remarks to the Author):

Most of my concerns have been satisfactorily addressed or acknowledged in the revised manuscript. As shown in the Author's Responses, the additional cause-specific analyses comparing other drugs with NVP and EFV actually showed significant benefits of NVP and EFV on non-AIDS-related mortality in the real-world setting. Should this be acknowledged or mentioned, the paper is acceptable.

Authors' reply:

Thank you for this positive feedback and valuable insight. We have acknowledged and discussed this important issue in the Discussion section in the revised manuscript (Line 291-295):

“It is noteworthy that, compared to the other less common third drugs, the non-AIDS-related mortality among PLHIV initiating NVP and EFV has significantly decreased. This may be related to the side effects of less common drugs, warranting further research to elucidate the underlying causes of this phenomenon.”

Reviewer #2 (Remarks to the Author):

The authors have adequately responded to all of my previous comments. I have no further comments.

Authors' reply:

Thank you for this positive feedback.